



# A Novel Approach to Assessing Nuisance Risk from Seismicity Induced by UK Shale Gas Development, with Implications for Future Policy Design

Gemma Cremen[1] and Maximilian J. Werner[2]

[1]Department of Civil, Environmental and Geomatic Engineering, University College London, London, UK.
[2]School of Earth Sciences, University of Bristol, Bristol, UK.

**Correspondence:** Gemma Cremen (g.cremen@ucl.ac.uk)

**Abstract.** We propose a novel framework for assessing the risk associated with seismicity induced from hydraulic fracturing, which has been a notable source of recent public concern. The framework combines statistical forecast models for injection-induced seismicity, ground motion prediction equations, and exposure models for affected areas, to quantitatively link the volume of fluid injected during operations with the potential for nuisance felt ground motions. Such (relatively small) motions

are expected to be more aligned with the public tolerance threshold for induced seismicity than larger ground shaking that could cause structural damage. This proactive type of framework, which facilitates control of the injection volume ahead of time for risk mitigation, has significant advantages over reactive-type magnitude and ground motion-based systems typically used for induced seismicity management. The framework is applied to the region surrounding the Preston New Road shale gas site in North West England. A notable finding is that the calculations are particularly sensitive to assumptions of the seismicity

forecast model used, i.e. whether it limits the cumulative seismic moment released for a given volume or assumes seismicity is consistent with the Gutenberg-Richter distribution for tectonic events. Finally, we discuss how the framework can be used to inform relevant policy.

## 1 Introduction

Awareness and concern regarding the impacts of seismicity induced by hydraulic fracturing has grown significantly in recent

years (e.g Williams et al., 2017; Davies et al., 2013; Whitmarsh et al., 2015; Ellsworth, 2013; Cotton et al., 2014), which may pose a threat to the future development of unconventional gas resources (Kraft et al., 2009). There is evidence that tolerance to such operations will be increased if the public is made aware of the potential consequences of the resulting ground shaking (Bommer et al., 2015; Giardini, 2009). In addition, understanding these consequences is critical for responsible decision-making by relevant political authorities (e.g. MacRae, 2006). It is therefore essential to develop methodologies for quantifying

and managing the hazard and risk posed by hydraulic-fracture induced seismicity.

Several hazard and risk assessment procedures have already been proposed in the literature for various types of induced seismicity. For example, Douglas and Aochi (2014) developed a conceptual model for assessing the risk of generating felt or damaging ground motions from enhanced geothermal systems (EGS), based on fluid injection rate. The model used information



on recent seismicity and ground shaking predictions from a ground motion prediction equation (GMPE) to obtain a real-time
hazard curve, which was combined with a fragility curve to quantify risk. Gupta and Baker (2019) developed a probabilistic
framework for estimating regional risk due to induced seismicity related to wastewater injection in Oklahoma that extends
conventional probabilistic seismic hazard analysis to account for spatiotemporally varying seismicity rates. Walters et al. (2015)
developed a qualitative risk assessment framework for triggered seismicity related to saltwater disposal and hydraulic fracturing
that included risk tolerance matrices to be considered by different stakeholders.

This paper proposes a novel risk assessment framework for hydraulic-fracture induced seismicity that directly links the
volume of fluid injected during an operation to its potential for causing nuisance ground motions, i.e. shaking that is an
inconvenience to society, and may raise annoyance or distress among the public (Foulger et al., 2018). This type of shaking is
expected to be more in line with public tolerances for induced seismicity than larger ground motions that have the potential
to cause structural damage (Bommer et al., 2015). The framework integrates, in a mathematically rigorous manner, statistical
forecast models for injection-induced seismicity, ground motion prediction equations for hydraulic fracturing, and exposure
models for nearby areas.

    The framework is applied to the region surrounding the Preston New Road (PNR) shale gas site in Lancashire, North West
England, where recent (2018/2019) hydraulic fracture operations resulted in 29 seismic events with local magnitude ($M_L$)
greater than 0 (e.g. Clarke et al., 2019), including eight that were felt by the local population (Baptie, 2019). We demonstrate
how the risk calculations can accommodate numerous styles of potential decision-making related to the regulation of hydraulic-
fracture-induced seismicity, and investigate the sensitivity of the calculations to certain model assumptions. The paper ends
with a discussion on ways in which the proposed framework could be used to design future policies related to the management
of hydraulic-fracture-induced risk in the UK.

## 2   Framework Outline

The proposed framework is a modified version of conventional probabilistic seismic hazard analysis (Cornell, 1968), where the
rate of earthquake occurence, the distribution of magnitudes, and therefore the rate of exceedance for a given intensity measure
($IM$), are conditioned on the total volume of fluid injected during a hydraulic fracture operation ($V_t$). It may be expressed as
follows:

$$\lambda(IM > x|V_t) = \sum_i^{n_s} \lambda(M_i > m_{min}|V_t) \left[ \int_{m_{min}}^{m_{max}|V_t} \int_0^{r_{max}} p(IM > x|m,r) f_{M_i|V_t}(m) f_{R_i}(r) \, dm \, dr \right] \tag{1}$$

where $n_s$ is the number of earthquake sources, $\lambda(a > b|c)$ is the rate at which $a$ exceeds $b$ given the occurrence of $c$, $M_i$
is the magnitude distribution for the $i$th source, $p(k|j)$ is the probability of $k$ given $j$, $m_{min}$ is the minimum magnitude
considered, $m_{max}|V_t$ is the maximum magnitude considered for a given injection volume, $f_Y(y)$ is the probability density
function of $Y$ evaluated at $y$, and $R_i$ is the distance distribution of distances from the $i$th source to the location of interest. The
"$\lambda(M_i > m_{min}|V_t)$" and "$f_{M_i|V_t}(m)$" terms are characterised by a statistical forecast model for injection-induced seismicity,
the "$p(IM > x|m,r)$" term is derived from ground shaking estimations by a GMPE designed for hydraulic fracturing events,




and the "$f_{R_i}(r)$" term is obtained from an exposure model of the affected region. While the framework is sufficiently flexible to cater for any intensity measure, we specifically use Peak Ground Velocity ($PGV$) as the measure of ground shaking in this study (i.e. $IM = PGV$ in equation 1) because of its close correlation with seismic intensity (Van Eck et al., 2006), and its ability to indicate damage for the small, shallow earthquakes of interest in this study (Bommer and Alarcon, 2006; Crowley

et al., 2018).

The framework is based on the assumption of a one-to-one relationship between the exceedance of tolerable ground shaking thresholds and nuisance risk, i.e.

$$p(NR_i | im > x_i) = 1 \qquad (2)$$

where $NR_i$ is the nuisance risk associated with the $i$th tolerable ground shaking threshold, $x_i$. Tolerance for potential ground

shaking may be dependent on the culture of those affected (Foulger et al., 2018), and a discussion with local stakeholders is therefore necessary to decide exactly what risk is acceptable (Giardini, 2009). However, our methodology provides a number of suggested tolerable ground shaking thresholds, based on previous studies associated with discomfort due to ground shaking (Bommer et al., 2006) and nuisance limits adopted for other types of vibration. These are: (1) $PGV = 0.9$ mm/s, which approximately corresponds with the velocity at which pile driving becomes 'barely perceptible' (Athanasopoulos and Pelekis, 2000);

(2) $PGV = 3$ mm/s, which is the velocity at which traffic-induced vibration becomes 'barely noticeable' (Barneich, 1985); (3) $PGV = 15$ mm/s, which is the lowest threshold of cosmetic damage for weak (i.e. unreinforced or light framed) structures, according to BSI (1993), and has been used in previous risk calculations for induced seismicity (Ader et al., 2019); and (4) $PGV = 50$ mm/s, which is the BSI (1993) threshold of cosmetic damage for strong (i.e. reinforced or framed) structures.

## 3 Case Study Framework Application

We apply the proposed framework to the region surrounding the Preston New Road (PNR) shale gas site in Lancashire, North West England, where hydraulic fracturing operations took place in late 2018 (at PNR-1z well) and mid 2019 (at PNR-2 well), resulting in a number of felt seismic events with maximum magnitude $M_L = 2.9$. For the purposes of this application, we assume that seismicity is produced from a point source 2 km deep (i.e. $n_s = 1$ in equation 1), at a respective latitude and longitude of 53.7873° North and 2.9511° West. This location corresponds to the approximate depth of the Bowland shale

targeted by the operation and the surface coordinates of the PNR-1z well, according to the 2018 hydraulic fracture plan of the operator (Cuadrilla, 2017).

### 3.1 Source and Ground Motion Modelling

We use the Hallo et al. (2014) injection-volume-based statistical model of event magnitudes, as it was used for real-time seismicity forecasting by the operator during hydraulic fracturing at PNR (Clarke et al., 2019). This model assumes that the

cumulative seismic moment released ($\sum M_o$) is related to the total volume of fluid injected ($V_t$) as follows:

$$\sum M_o = S_{EFF} \mu V_t \qquad (3)$$




where $\mu$ is the rock shear modulus. $S_{EFF}$ depends on the medium type and the nature of the injected material, and represents the ratio of $\sum M_o$ to its theoretical maximum ($\mu V_t$) if there was no aseismic deformation (Hallo et al., 2012). For this formulation:

$$\sum_{j}^{n} M_{jo} \leq \sum M_o \qquad (4)$$

where $M_{jo}$ is the seismic moment equivalent of the $j$th earthquake and $n$ is the number of earthquakes that occur, which is a random variable that follows a Poisson probability mass function with mean $N = \sum_{i}^{n_s} \lambda(M_i > m_{min}|V_t)$ from equation 1. $m_{max}|V_t$ in equation 1 for the $i$th event is defined as:

$$m_{max,i}|V_t = \alpha_{i-1} \sum M_{o,w} \qquad (5)$$

where $\alpha_{i-1}$ is the fraction of $\sum M_{o,w}$, the total seismic moment in moment magnitude terms, still to be released after the occurrence of the $(i-1)$th event.

$\mu$ is assumed to be 20 GPa throughout this study, from previous work on PNR seismicity (Clarke et al., 2019). The sets of $S_{EFF}$, $m_{min}$ and $b$- values used are those fit by Clarke et al. (2019) for seismicity produced during PNR-1z operations, where 17 sleeves were stimulated with a total injection volume of approximately 4200 m$^3$ (see Table 1). We treat the stimulated sleeves before Sleeve 18 (i.e. Sleeves 1, 2, 3, 12, 13, and 14) as independent, and use the relevant set of sleeve-specific seismicity parameters for each. For the 11 remaining stimulated sleeves, we use the set of seismicity parameters fit over their cumulative injected volume, as they were found to intersect the same fault (Clarke et al., 2019). It should be noted that some sets of seismicity parameters used were fit using a mixture of event magnitudes reported on moment and local scales (Clarke et al., 2019), yet the size of forecasted events are always measured on the moment magnitude scale. This discrepancy is deemed acceptable however, given that the precise relationship between the scales is yet to be established (Mancini et al., 2019).

**Table 1.** Seismicity parameters used in this study, from Clarke et al. (2019).

| Sleeve Number | $\log_{10}(S_{EFF})$ | $m_{min}$ | $b$- value |
|---|---|---|---|
| 1 | -2.61 | -1.1 | 1.85 |
| 2 | -3.42 | -1.3 | 2.08 |
| 3 | -2.81 | -1.4 | 1.30 |
| 12 | -2.23 | -2.3 | 1.06 |
| 13 | -3.13 | -2.2 | 1.07 |
| 14 | -2.39 | -2.3 | 1.24 |
| 18-41 | -1.90 | -0.6 | 1.22 |

Ground shaking ($p(IM > x|m,r)$ in equation 1) is predicted using the ground motion prediction equation of Cremen et al. (2019b), which was specifically designed for hydraulic-fracture induced seismicity in the UK. This equation characterises ground motion intensity in terms of moment magnitude and hypocentral distance at the location of interest. It is intended to model ground motion amplitudes for events with $M_w < 3$ at hypocentral distances < 6 km.





## 3.2 Exposure Database

110

The considered exposure database ($f_R(r)$ in equation 1) comprises buildings located within a 5 km hypocentral distance of the event location (Figure 1). Building data are obtained from Ordnance Survey (OS) mapping, accessed through the Edina Digimap service (Morris et al., 2000). Building footprint information is acquired from the 'Buildings' layer of the OS VectorMap Local product, and building height information is acquired from the OS MasterMap 'Building Height Attribute'

115 database. Height and footprint data are matched via their geographic coordinates; we consider the corresponding building height for a given building footprint to be the closest located within 10 m. To exclude small non-habitable structures, we neglect buildings with footprint areas $< 40$ m$^2$ and/or known building heights $< 3.5$ m. This results in a final exposure database of 4195 buildings.

We also separately consider important buildings, or critical sites in which the occupants (or equipment) may be more sensitive

120 to the effects of vibrations from ground shaking than those of conventional residential or commercial buildings (Ader et al., 2019; Walters et al., 2015). We exclusively focus on educational and medical facilities within 5 km of the event, which are identified from the 'Important Buildings' layer of the OS VectorMap Local. We neglect all important buildings with footprint areas $< 100$ m$^2$, which is the typical size of a classroom (Haylock, 2001). This results in a final database of six important buildings.

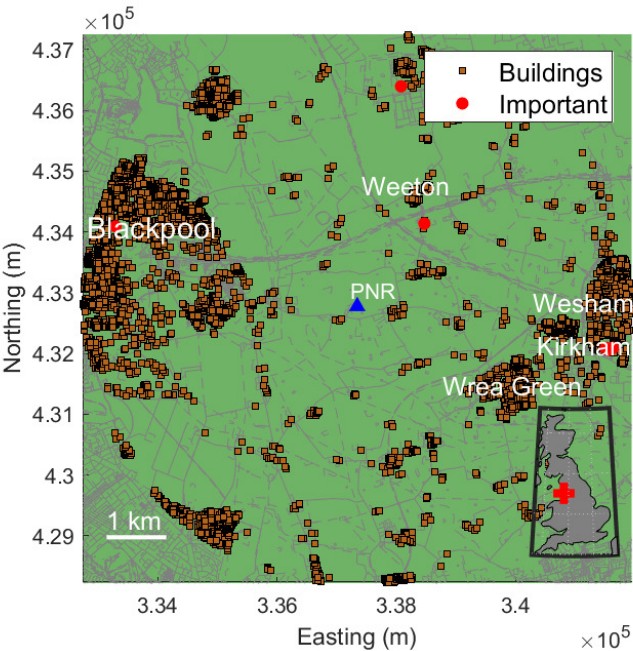

**Figure 1.** All buildings and important (i.e. educational and medical) buildings considered within 5 km hypocentral distance of the case study event location at the Preston New Road (PNR) hydraulic fracture site (inset highlights location relative to all of Great Britain). © OpenStreetMap contributors 2019 (*https://www.openstreetmap.org/copyright*).



## 3.3 Monte Carlo Sampling Procedure

Equation 1 describes ground motion exceedance at a single site. To capture the risk across the multiple buildings of interest in this study and correctly account for ground motion variability (Bourne et al., 2015), we employ a Monte Carlo sampling approach (e.g. Assatourians and Atkinson, 2013; Musson, 1999). This procedure involves the following steps for a given injection volume:

1. Calculate the corresponding total seismic moment, using equation 3.

2. Choose a single random event from the magnitude distribution $f_{M|V_t}(m)$ of equation 1, which is truncated on the left by $m_{min}$ and on the right by $m_{max,i}|V_t$ (as defined in equation 5).

3. Use the Cremen et al. (2019b) GMPE to simulate a random inter-event variability and random intra-event variabilities for each site.

4. Calculate median ground motion predictions from the GMPE at each site for the given combination of $\{m,r\}$, and add the inter- and intra-event variabilities generated in step 3 to simulate ground motion intensities.

5. Repeat Step 2-4 until the total seismic moment of the sampled events is equal to that calculated in step 1 to within a small tolerance.

6. Repeat Steps 2-5 1000 times to generate 1000 potential catalogs corresponding to the given injected volume.

## 3.4 Modelling Validation

The proposed risk modelling approach is validated using data observed during the 2018 hydraulic fracturing operations at the PNR 1-z well. We complete the Monte Carlo sampling procedure for the actual volumes of fluid injected during those operations, using the UK Oil and Gas Authority's database on PNR operations (https://www.ogauthority.co.uk/onshore/onshore-reports-and-data/preston-new-road-pnr-1z-hydraulic- fracturing-operations-data/). Figures 2a-2c compare the predicted numbers of earthquakes with those observed across selected sleeves (similar results are obtained for the remaining sleeves). It is seen that the observations almost always lie within the 1st and 99th percentile predictions of the model, thus we can conclude that the source model used is appropriate for forecasting the seismicity of interest.

Figure 2d compares ground shaking predictions at locations in the exposure model with observed ground motion amplitudes, for all forecasted/recorded events of $M >\approx 0.1$, within a 2-3 km hypocentral distance range. This distance range is chosen since it corresponds to: (1) a similar number of predictions (229 per synthetic catalog) and observations (173); and (2) an almost identical mean hypocentral distance for predictions (2.66 km per synthetic catalog) and observations (2.69 km). It is seen that the observed ground shaking amplitudes generally lie within the 1st and 99th percentile predictions, and therefore it is clear that the proposed model is adequately capturing the shaking intensity (risk) of interest. This confirms that the slight magnitude scale discrepancy in the source model (see Section 3.1) does not inhibit the overall performance of the calculations.

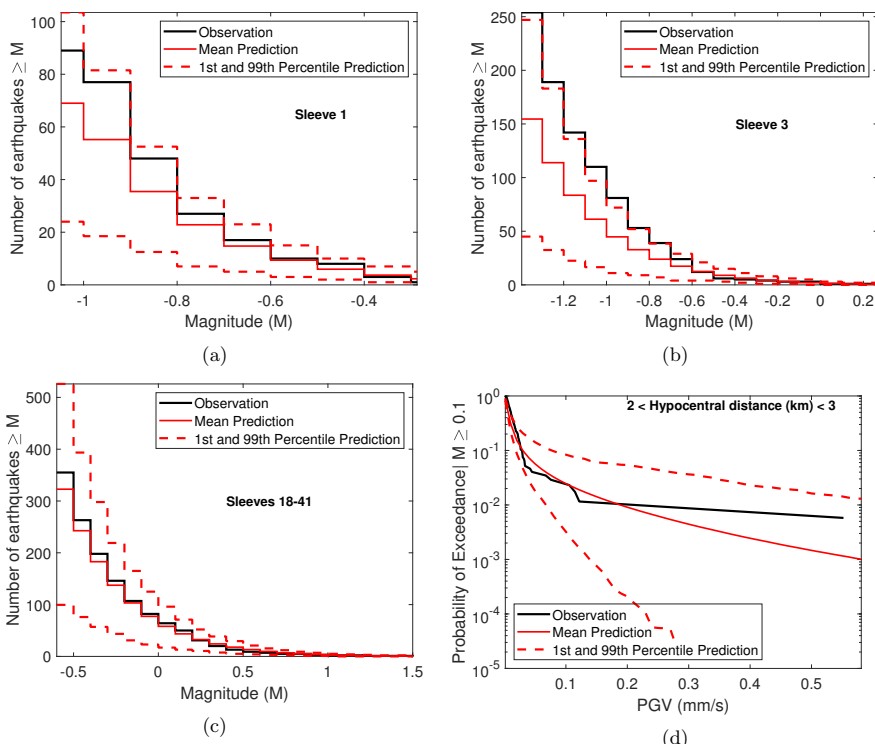

**Figure 2.** Validating the risk modelling approach of this study, using data observed during hydraulic fracturing of the PNR 1-z well: (a)-(c) Comparing forecasted numbers of earthquakes with those observed; and (d) Comparing predicted ground shaking with observed ground motion amplitudes.

## 4  Case Study Results

### 4.1  Magnitude-Specific Calculations

We first examine the risk associated with the occurrence of specific moment magnitudes ($M_w$) up to $M_w = 3$, which is the maximum applicable magnitude for the Cremen et al. (2019b) GMPE. We repeat steps 3 and 4 of Section 3.3 1000 times for $M_w$ between 0.1 and 3, in increments of 0.1. Results of the calculations are found in Figure 3, where they are presented three different ways to accommodate various potential styles of decision-making. Figures 3a-3d summarise the probability of exceeding the prescribed risk thresholds at least once across different magnitude-distance bins, considering all buildings. As expected, the probability of exceeding the thresholds increases for increasing magnitude and decreasing hypocentral distance. It is observed that the $PGV = 0.9$ mm/s (pile driving perceptibility) threshold exceedance probability becomes non-zero for $M_w > \approx 1.1$ at close distances, and for $M_w > \approx 1.8$ at all examined distances. The $PGV = 3$ mm/s (traffic noticeable) threshold exceedance probability becomes non-zero for $M_w > \approx 1.6$, and non-zero at all examined distances for $M_w > \approx 2.3$. The $PGV = 15$ mm/s (cosmetic damage for weak structures) threshold exceedance probability becomes non-zero for $M_w > \approx$


2.1, and for $M_w >\approx 2.8$ at all distances of interest . The $PGV = 50$ mm/s (cosmetic damage for strong structures) threshold exceedance probability becomes non-zero for $M_w >\approx 2.5$, but does not become non-zero across all examined distances for any magnitude of interest in this study.

Figure 3e and Figure 3f examine the risk associated with three specific magnitudes: (1) $M_L = 0.5$, which is the current red light ("stop injection") threshold for hydraulic-fracture-induced seismicity in the UK, (2) $M_L = 2.1$, which was the second-largest event that occurred during 2018/2019 PNR operations, and (3) $M_L = 2.9$, which was the largest event that occurred during 2018/2019 PNR operations. These magnitudes are converted to moment magnitude for input to the Cremen et al. (2019b) GMPE, using the empirical relationship derived by Butcher et al. (2019) for small magnitudes in a similar geologic 175    setting; this is an approximate conversion, since the relationship between the scales is uncertain (Mancini et al., 2019). For this relationship, (1) $M_L = 0.5$ is equivalent to $M_w = 1.1$, (2) $M_L = 2.1$ is equivalent to $M_w = 2.2$, and (3) $M_L = 2.9$ is equivalent to $M_w = 2.7$.

Figure 3e shows the probability of exceeding different $PGV$ levels at least once, across all considered buildings (magenta curves) and important buildings (blue curves). It is seen that the current red light event for UK hydraulic fracturing has only 180    a negligible ($\approx 0.2\%$) probability of exceeding the lowest of the four considered tolerable risk thresholds at the location of at least one building in the exposure model. An event equivalent in size to the second largest 2019 event will almost certainly exceed both the pile driving and traffic thresholds, and has a 2% chance of causing cosmetic damage in a worst case scenario (i.e. weak structure). An event equivalent in size to the largest 2019 event exceeds the first three considered tolerable risk thresholds with certainty, and there is an approximate 10% chance that it will result in ground motions that cause cosmetic 185    damage of at least one building in a best case scenario (i.e. strong structure). The predicted occurrence of cosmetic damage for $M_w = 2.7$ is consistent with actual observations (despite the hypothetical event being located approximately 0.8 km to the west of where the actual event occurred, at a 0.5 km shallower depth); the British Geological Survey assigned the event an intensity of 6 on the European Macroseismic Intensity scale (Grünthal, 1998), meaning "slightly damaging", based on data from more than 2000 felt reports (British Geological Survey, 2019).

It is also seen in Figure 3e that the curves associated with important buildings are positioned to the left of those associated with all buildings, for the same magnitude event. This implies that the risk for important buildings is lower than that for all buildings. For example in the worst case scenario, there is only approximately 10% probability of cosmetic damage occurring in at least one important building versus near certainty of this type of damage occurring in at least one building, for an event equivalent to the largest that occurred in 2019. The smaller risk associated with important buildings makes sense, since they 195    are located further away from the hydraulic fracture site than the closest of all considered buildings (see Figure 1).

Figure 3f shows the average number of all buildings (magenta curves) and important buildings (blue curves) at which different $PGV$ levels are exceeded for the three specific magnitudes examined. Less than one building is expected to experience shaking that exceeds the lowest of the four considered tolerable risk thresholds, for an event equal in size to the current UK red light event. Approximately 30 buildings are expected to experience ground motions that exceed the traffic threshold for an 200    event equivalent in size to the second largest that occurred in 2019, and approximately 20 buildings are expected to experience cosmetic damage in a worst case scenario for an event equivalent in size to the largest in 2019. In the case of important




buildings, less than one is expected to experience exceedance of the pile driving threshold for either $M_w = 1.1$ or $M_w = 2.2$, and approximately one is expected to experience exceedance of the traffic threshold for $M_w = 2.7$.

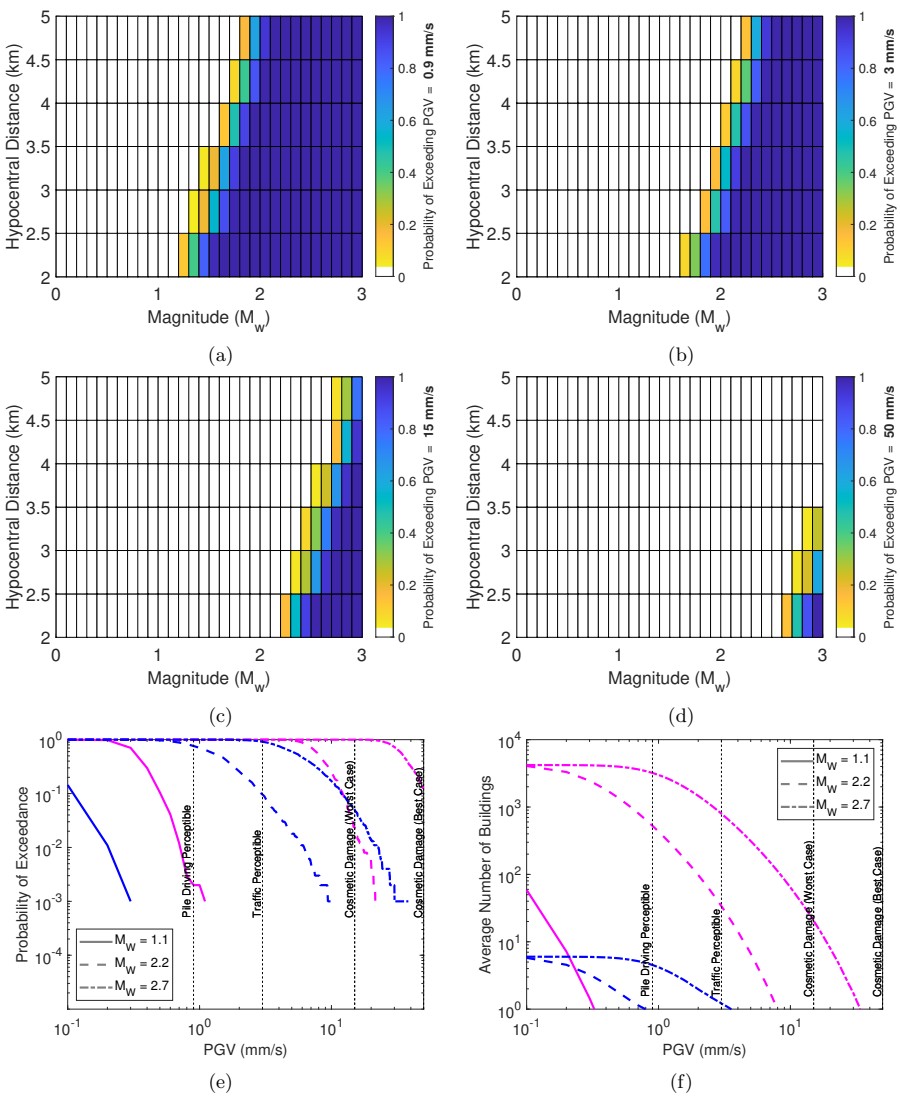

**Figure 3.** Magnitude-specific risk calculations: (a)-(d) summarise the probability of $PGV$ exceeding the prescribed risk thresholds (0.9 mm/s, 3 mm/s, 15 mm/s, and 50 mm/s respectively) at least once across different magnitude-distance bins; (e) highlights, for three specific magnitudes, the probability of exceeding various $PGV$ levels at at least one building (magenta curves) and one important building (blue curves); and (f) shows, for three specific magnitudes, the average number of buildings (blue curves) and important buildings (magenta curves) at which various $PGV$ levels are exceeded.





## 4.2 Injection-Volume-Based Calculations

We use the procedure outlined in Section 3.3 to examine the risk associated with the following injection volumes: 500 m$^3$, 1000 m$^3$, 5000 m$^3$, 10,000 m$^3$, 15,000 m$^3$, 20,000 m$^3$, 30,000 m$^3$, 40,000 m$^3$, and 50,000 m$^3$. These values capture the typical range of injection volumes planned/used for hydraulic fracturing operations in both the UK (Westaway, 2016; Cuadrilla, 2017, 2019; Third Energy, 2017; Clarke et al., 2019) and North America (Johnson and Johnson, 2012; Precht and Dempster, 2012; Ground Water Protection Council, 2009; Gallegos et al., 2015). We assume that each volume is divided evenly among the 17

stimulated sleeves of the PNR-1 operation, and simulate seismicity according to the sleeve-dependent parameters discussed in Section 3. Results of the calculations are summarised in Figure 4, using similar presentation methods as those introduced in Section 4.1.

Figures 4a-4d show the probability of exceeding the prescribed risk thresholds at least once across different volume-distance bins, considering all buildings. The probability of exceeding the thresholds clearly increases as injection volume increases

and hypocentral distance decreases, in line with expectations. It is seen that the $PGV = 0.9$ mm/s (pile driving) threshold exceedance probability becomes non-zero at close distances for 1000 m$^3$ of injected volume, and at all examined distances for 10,000 m$^3$. The $PGV = 3$ mm/s (traffic) threshold exceedance probability becomes non-zero for 5000 m$^3$, and non-zero at all examined distances for 40,000 m$^3$. The $PGV = 15$ mm/s (cosmetic damage for weak structures) threshold exceedance probability becomes non-zero for 40,000 m$^3$, but does not become non-zero across all examined distances for any injection

volume of interest. The $PGV = 50$ mm/s (cosmetic damage for strong structures) threshold is not exceeded for the examined injection volumes.

Figure 4e and Figure 4f examine the risk associated with the specific injection volumes of interest, across different $PGV$ levels. Figure 4e shows the probability of exceeding a given value of $PGV$ at least once, across all considered buildings (magenta curves) and important buildings (blue curves). It is seen that there is no chance of exceeding any of the considered

tolerable risk thresholds for 500 m$^3$ injected volume, and there is only a negligible ($\approx$ 1%) probability of exceeding the lowest of the four considered thresholds at least once for 1000 m$^3$. 5000 m$^3$, 10,000 m$^3$, 15,000 m$^3$, 20,000 m$^3$, and 30,000 m$^3$ have approximately 2%, 10%, 30%, 50%, and 80% chance respectively, of generating ground motions that exceed the traffic threshold at the location of at least one building in the exposure model. The largest two injection volumes considered (i.e. 40,000 m$^3$ and 50,000 m$^3$) will almost certainly result in shaking that exceeds the traffic threshold, but will only result in

cosmetic damage in a worst case scenario (weak structure) with less than 10% probability. It appears that no injected volumes examined have any chance of causing cosmetic damage in a best case scenario (strong structure) (preliminary calculations suggest that approximately 80,000 m$^3$ is required for a non-zero probability of this damage occurring). Curves associated with important buildings are positioned to the left of those associated with all buildings in Figure 4e, implying lower risk for important buildings as discussed in Section 4.1. For example in the worst case scenario, there is negligible ($\approx$ 1%) chance

of cosmetic damage occurring in at least one important building versus approximately 9% probability of this type of damage occurring in at least one building, for the largest considered injected volume.





Figure 4f shows the average number of all buildings (magenta curves) and important buildings (blue curves) at which different $PGV$ levels are exceeded for the injection volumes examined. Less than one important building is expected to experience shaking that exceeds the lowest of the four considered tolerable risk thresholds for any injected volume analysed,

and less than one building of any type is expected to experience exceedance of this threshold for both 500 m$^3$ and 1000 m$^3$ injected volumes. Less than 10 buildings are expected to experience exceedance of the lowest threshold for 5000 m$^3$, and between 10 and 100 buildings are expected to experience shaking above this threshold for 10,000 m$^3$, 15,000 m$^3$, and 20,000 m$^3$. Between 10 and 100 buildings are expected to experience exceedance of the traffic threshold for the 30,000 m$^3$, 40,000 m$^3$, and 50,000 m$^3$. Less than one building is expected to experience cosmetic damage in a worst case scenario, for any injected

volume examined.

We disaggregate the results shown in Figure 4 and examine the risk associated with each injection volume in terms of the frequency of risk threshold exceedances attributable to different event magnitudes (Bazzurro and Allin Cornell, 1999). Frequencies of exceedance are important to consider, given that the number of shaking episodes people experience is expected to directly influence their tolerance limit (Bommer et al., 2006); a single occurrence of ground shaking with relatively high

intensity may be significantly less concerning to local populations than a continuous series of weaker felt ground motions (Bourne et al., 2015).

We provide the results of the disaggregation in Figure 5 for the $PGV = 0.9$ mm/s and the $PGV = 3$ mm/s thresholds. These plots show, for each injected volume of interest across different magnitudes, the average number of events per catalog associated with ground shaking that exceeds the risk threshold of interest at the closest building (or important building) in

the exposure model to the location of seismicity. It is clear that the largest contributor to exceeding either threshold is not always the maximum magnitude experienced, particularly for larger volumes of injected fluid. For example, for all considered buildings and an injected volume of 50,000 m$^3$, $M_w$ 2.2 events result in approximately four times the number of traffic threshold exceedances than $M_w$ 2.5 earthquakes. Intermediate magnitudes govern the hazard and risk because they occur more frequently than larger magnitudes. In line with previous studies (Bourne et al., 2015), the findings of the disaggregation underline the fact

that it is not always useful to focus exclusively on the maximum magnitude parameter (e.g. McGarr, 2014) when assessing the risk due to induced seismicity.

## 5 Exploring the Impacts of Modelling Assumptions

The application of the proposed risk framework presented in Sections 3 and 4 made use of a number of assumptions related to the source modelling and prediction of ground motion. For example, we assumed that all seismicity was co-located and

that there were no spatial correlations in the ground motions from a given event. Alternative assumptions may also be valid however, depending on the level of information available at the time a risk analysis is being conducted. This section explores the impact of some of these assumptions on the risk calculations. We first discuss the alternative assumptions of interest, and present the results of their impact for all considered buildings in the exposure model across 5000 m$^3$, 20,000 m$^3$, and 50,000 m$^3$ of injected fluid in Figure 9. In all cases, injection volume is divided evenly between sleeves as in Section 4.2.


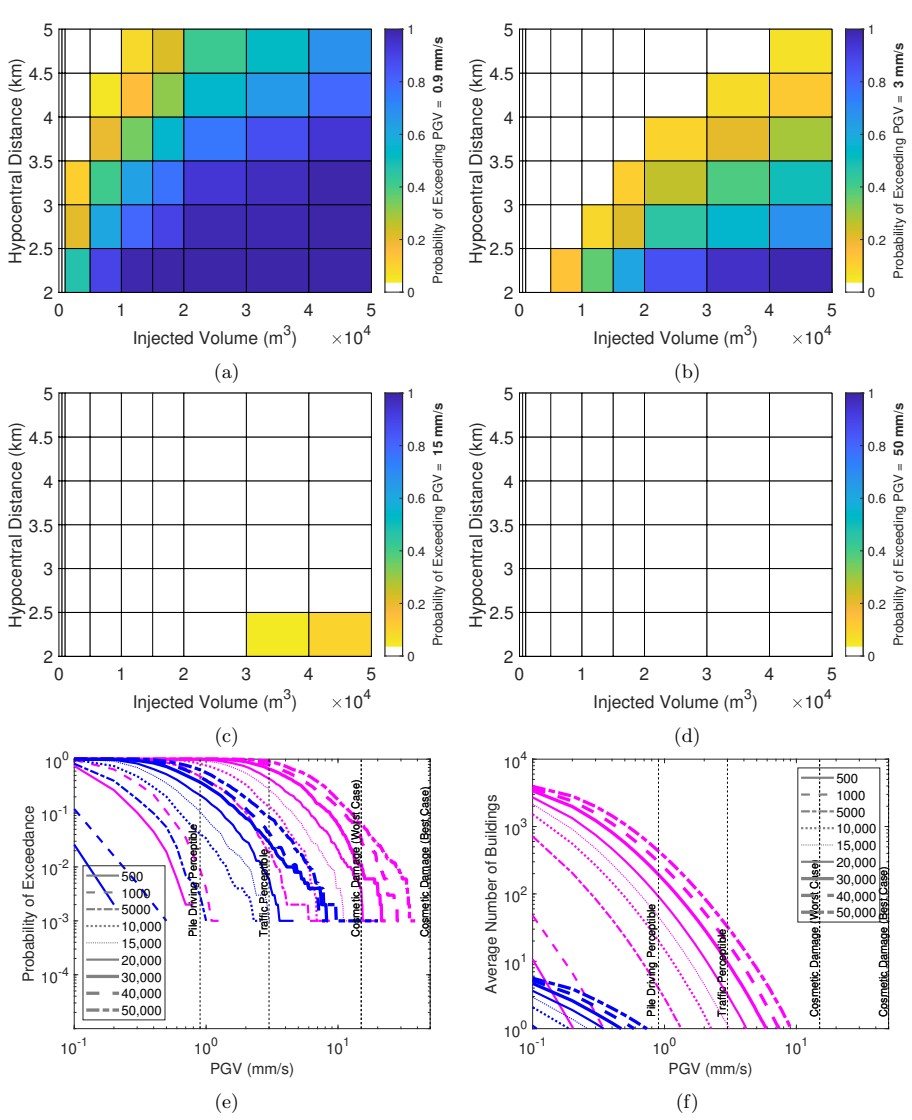

**Figure 4.** Injection-volume-based risk calculations: (a) - (d) summarise the probability of $PGV$ exceeding the prescribed risk thresholds (0.9 mm/s, 3 mm/s, 15 mm/s, and 50 mm/s respectively) across different volume-distance bins; (e) highlights, for specific volumes, the probability of exceeding various $PGV$ levels at at least one building (magenta curves) and one important building (blue curves); and (f) shows, for specific volumes, the average number of buildings (magenta curves) and important buildings (blue curves) at which various $PGV$ levels are exceeded.



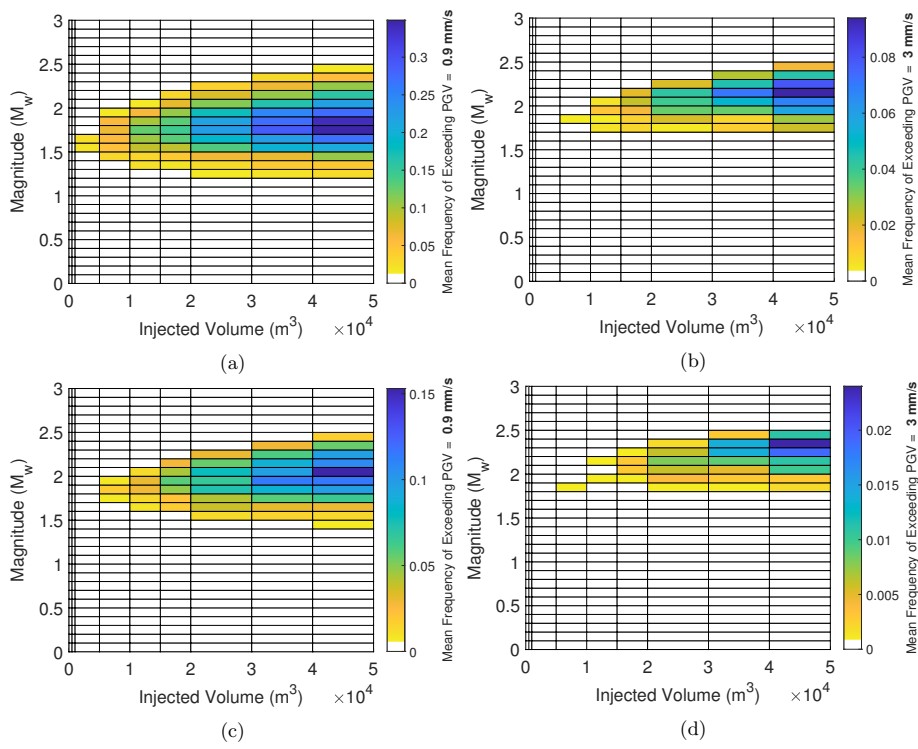

**Figure 5.** Ground motion disaggregation for $PGV = 0.9$ mm/s (left panels) and $PGV = 3$ mm/s (right panels) risk thresholds, across (a)-(b) all buildings and (c)-(d) important buildings.

## 5.1 Uncertain Seismicity Parameters

The analysis conducted in Section 4.2 relied on after-the-fact observation-based estimates of the seismicity parameters $b$ and $S_{EFF}$. In reality however, there is likely to be a large degree of uncertainty in these parameters before operations are carried out (e.g. Bommer et al., 2015; Mignan et al., 2017), when we expect the proposed risk assessment procedure to be at its most valuable in forecasting nuisance risk associated with planned injection volumes. It is therefore important to understand how uncertainties in these parameters affect the calculations of the framework. To do this, we conduct a hypothetical *a priori* risk assessment, in which reasonable uncertainties in the seismicity parameters are introduced.

We assume a uniform distribution of $b$-values values between 1 and 2. This is a sensible approach, given that the bounds of the distribution approximately represent the two opposite conditions of fault reactivation within a seismically active region (i.e. $b = 1$) and hydraulic fracture interaction with natural fractures (i.e. $b = 2$) (Eaton et al., 2014; Chen et al., 2018). We assume a uniform distribution of $\log_{10}(S_{EFF})$ values between -4 and -1. The chosen bounds are within the limits of those expected for hydraulic fracturing and enhanced geothermal systems (Hallo et al., 2014), which the values for hydraulic fracturing may reach





in extreme cases (Hallo et al., 2012). They are also in line with the range of values observed during the hydraulic fracturing operation studied in Verdon and Budge (2018).

Uncertainty in the seismicity parameters affects Steps 1, 2, and 6 of the Monte Carlo sampling procedure (Section 3.3). In
Step 1, single random $S_{EFF}$ and $b$- values are chosen from their respective distributions. The chosen $S_{EFF}$ value is used to calculate the total seismic moment for the given injection volume, and the sampled $b$-value is used to characterise the $f_{M|V_t}(m)$ distribution used in Step 2. In Step 6, Steps 1-5 are instead repeated, such that each synthetic catalog is generated from different seismicity parameter values.

## 5.2 Uncertain Event Locations

We have thus far assumed that the location of seismicity is known and that all events are co-located in space. In realistic scenarios however, there will be some uncertainty on event locations (e.g. Bao and Eaton, 2016; Verdon et al., 2019). To explore the impact of this uncertainty on the risk calculations, we repeat the analyses for random event locations.

We assume events are produced from point sources that are uniformly distributed within 1.4 km of the surface coordinates of the PNR-1z well, where 1.4 km corresponds to the lateral length of the PNR 1-z well (1 km) (Cuadrilla, 2017) plus the distance
of the largest event from the injection point (400 m) during previous hydraulic fracturing at a nearby shale gas site (Clarke et al., 2014). (The maximum distance between the surface coordinates of the PNR-1z well and the 57 events detected by the BGS surface array during 2018 hydraulic fracture operations was 1.38 km, which further confirms that 1.4 km is a sensible distance cap to assume). A uniform distribution is chosen to accommodate hypothetical cases in which the direction of the intended well path is still unknown.

Event depths are assumed to be uniformly distributed between 1.5 km and 3 km. This is a reasonable approach, given that the bounds of the distribution approximately correspond to the maximum and minimum depths of shale at the PNR site (Cuadrilla, 2017). Note that the Cremen et al. (2019b) GMPE is not strictly intended for hypocentral distances $< 2$ km, however we deem its use in such cases acceptable here in the absence of an appropriate alternative model that has been calibrated for such shallow depths and given that these distances comprise less than 0.3% of all those simulated within 5 km.

Accounting for event location uncertainty requires an additional task in the Section 3.3 Monte Carlo risk procedure between Steps 1 and 2, in which single distance-to-well and depth values are sampled from their respective assumed distributions to generate a random earthquake location. The exposure database examined also varies for each event; assessed buildings are selected according to the same hypocentral distance, height, and footprint criteria discussed in Section 3.2.

## 5.3 Different Rupture Behaviour

The statistical earthquake forecast model used in our analysis assumed a deterministic limit on earthquake magnitudes, based on the volume of fluid injected (see equation 5). While this model performed well for forecasting events during 2018 operations at PNR (see Section 3.4; Clarke et al., 2019) and closely aligned with observations when applied in a pseudoprospective manner for hydraulic fracture operations at Horn River in Canada (Verdon and Budge, 2018), there is ample evidence in the literature





to suggest that earthquakes may not be limited in size by the associated industrial activity (e.g Atkinson et al., 2016; Gischig, 2015; Mignan et al., 2019; Lee et al., 2019).

We now test the implications on the risk calculations of instead using the Van der Elst et al. (2016) source model for simulating events during fluid injection, which assumes that the largest magnitudes that occur for a given injection volume are consistent with the sampling statistics of the Gutenberg-Richter distribution for tectonic events. This approach was found to correspond well with magnitude-volume observations for recent fluid induced seismicity during a geothermal stimulation in Finland (Kwiatek et al., 2019).

The Van der Elst et al. (2016) model is based on the Shapiro et al. (2010) seismogenic-index ($S_i$) equation, which is expressed as follows:

$$\log_{10} N = S_i + \log_{10}(V_t) - bm \tag{6}$$

where $N$ is the expected number of earthquakes larger than magnitude $m$ that occur due to injected fluid volume $V_t$ and the $S_i$ parameter depends on the seismotectonic features of the region of interest. For this modelling approach, the "$\lambda(M > m_{min}|V_t)$" term of equation 1 becomes

$$\lambda(M > m_{min}|V_t) = V_t(10^{S_i - bm_{min}}) \tag{7}$$

In the following calculations, we use the $S_i$ parameters that correspond to the sets of $S_{EFF}$, $m_{min}$, and $b$- values detailed in Section 3.1. All other parameters used are as defined previously. We first test the validity of the model (Figure 6), using observed seismicity data for 2018 PNR fracture operations as outlined in Section 3.4. The observed data generally lie within the 1st and 99th percentile bounds of model predictions for both seismicity (similar results are found for sleeves not shown) and ground shaking. We thus conclude that the Van der Elst et al. (2016) model is a reasonable alternative assumption about rupture behaviour for this case.

This assumption affects all steps of the Monte Carlo procedure in Section 3.3. In Step 1 (which is now also repeated in Step 6), the total number of earthquakes above $m_{min}$ is simulated from a Poisson probability mass function with mean $N$ (equation 6), which is the number of times Steps 2-4 are repeated in Step 5. In Step 2, $m_{max}|V_t = 6.5$, which is the most likely maximum magnitude of UK tectonic earthquakes (Woessner et al., 2015). Steps 3 and 4 remain the same as in Section 3.3 if the sampled magnitude is less than 3, which is the maximum applicable magnitude for the Cremen et al. (2019b) GMPE. The Douglas et al. (2013) Model 1 is used for $M_w > 3$, as it was fit using some data above this magnitude and was found to be a promising candidate model for predicting ground motions due to UK hydraulic-fracture-induced seismicity (e.g. Cremen et al., 2019b). It does overestimate variability in these ground motions however (Cremen et al., 2019a), so we adjust the inter- and intra-event standard deviations of the model for this case, through mixed-effects regression of the data used to fit the Cremen et al. (2019b) GMPE (Scasserra et al., 2009). The modified values of inter- and intra-event variability used for the Douglas et al. (2013) equation are 0.337 and 0.778 respectively.

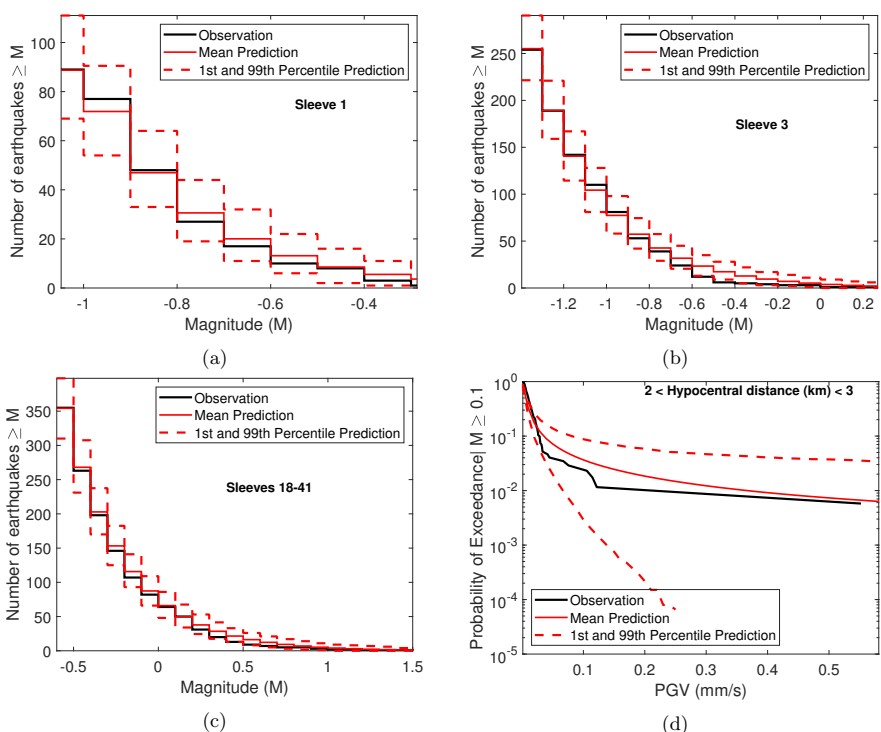

**Figure 6.** Validating the Van der Elst et al. (2016) approach as an alternative assumption about rupture behaviour for the case study: (a)-(c) Comparing forecasted numbers of earthquakes with those observed; and (d) Comparing predicted ground shaking with observed ground motion amplitudes.

## 5.4 Spatial Correlation in Ground Motions

The modelling approach of Section 3.3 assumes that the ground motion intensities generated at each site by a given event are independent. However, it is well documented that this assumption may not be valid if the sites are located closely in space (e.g. Boore et al., 2003; Wang and Takada, 2005), and neglecting spatial dependency in ground motion amplitudes may have a notable impact on the corresponding hazard and risk calculations (e.g. Park et al., 2007).

To understand the effect of accounting for ground motion spatial correlation in our risk calculations, we implement the model of Esposito and Iervolino (2012) for $PGV$, which accounts for dependencies in the corresponding intra-event term ($\epsilon$) of the GMPE at $n$ locations in space. Intra-event variabilities are sampled from a multivariate normal distribution, following



Jayaram and Baker (2008), with 0 mean and the following covariance matrix:

$$
\Sigma_\epsilon = \sigma_{intra}^2
\begin{bmatrix}
1 & \rho(h_{12}) & \ldots & \rho(h_{1n}) \\
\rho(h_{21}) & 1 & \ldots & \rho(h_{2n}) \\
\vdots & \vdots & \ddots & \vdots \\
\rho(h_{n1}) & \rho(h_{n2}) & \ldots & 1
\end{bmatrix}
\tag{8}
$$

where $\Sigma_\epsilon$ denotes the covariance matrix of $\epsilon$, $\sigma_{intra}$ is the intra-event standard deviation from the GMPE and $\rho(x_{i,j})$ is the correlation between the $i$th and $j$th $PGV$ intra-event residuals separated by distance $x$ km, defined as follows:

$$
\rho(x_{i,j}) = e^{-3x/13.7}
\tag{9}
$$

We test the validity of assuming the above spatial correlation model in our ground motion predictions (Figure 7), using observed ground motion data for 2018 PNR fracture operations as outlined in Section 3.4. The observed data generally lie within the 1st
and 99th percentile bounds of model predictions that account for spatial correlation, and we thus conclude that the Esposito and Iervolino (2012) model is a reasonable alternative assumption about ground motion spatial dependence for this case. The introduction of spatial correlation affects Step 3 of the Section 3.3 Monte Carlo procedure, where intra-event variabilities are instead sampled using the zero-mean multivariate normal distribution defined in equations 8 and 9 .

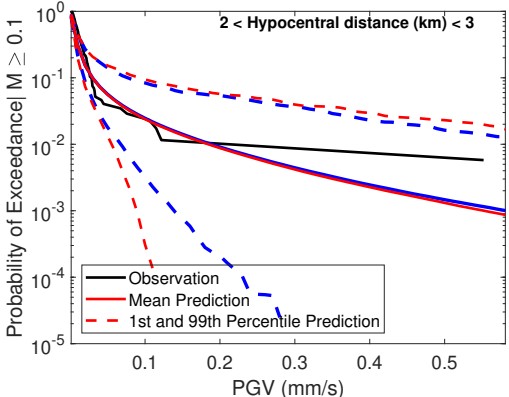

**Figure 7.** Validating the assumption of spatial correlation in the ground motions of the case study: Comparing predicted ground shaking with observed ground motion amplitudes. Also show in blue are the predictions for the base (uncorrelated) case.

### 5.5 Different GMPE

The Cremen et al. (2019b) GMPE used in our assessment was specifically fit using data observed during both hydraulic fracturing operations at PNR, and is therefore a particularly appropriate choice of ground motion model for the risk calculations of interest. On the other hand, pre-planning hazard calculations for PNR (Arup, 2014) employed the hypocentral distance model of Akkar et al. (2014) for predicting ground motion amplitudes, which is intended for application to crustal seismicity with much larger magnitudes than those that actually occurred at the site ($M_w > 4$). As expected for extrapolation of a GMPE to





smaller magnitudes (e.g., Bommer et al., 2007), this equation was consequently found to significantly overpredict the resulting ground motions (Cremen et al., 2019a, b). We examine the effect on the risk calculations of using this GMPE instead, which requires the Cremen et al. (2019b) GMPE to be substituted for the Akkar et al. (2014) equation in Steps 3 and 4 of the procedure in Section 3.3. We assume a $V_{s30}$ value of 200 m/s for the GMPE, in line with Arup (2014).

### 5.6  Impacts of Modelling Assumptions

Figure 9 highlights the impact of the alternative modelling assumptions on the probability of exceeding $PGV$ at least once in the exposure model, for specific volumes of injected fluid. The right-hand panel of the figure (b,d, and f) present the ratio of the risk curves in the corresponding left-hand panel. For a given value of $PGV$, this ratio may be calculated as follows:

$$\text{Ratio of Risk Curve } i = \frac{Q_i}{Q_b} \tag{10}$$

where $Q_i$ is the value of the risk curve for the $i$th alternative modelling assumption and $Q_b$ is its value in the base case (i.e.
the original modelling approach outlined in Section 3). The stability of the results in both plots of Figure 9 were assessed by simulating 100 bootstrap samples of the event catalogs and their associated $PGV$ values, and then re-computing the values on the y-axes. Transparent regions of the curves indicate low stability, where the bootstrapped y-values have coefficients of variation greater than 0.5. The following discussion focuses on the stable portions of the results.

The effect of the assumptions varies across different tolerable risk thresholds and volumes of injected fluid. Assuming the
Van der Elst et al. (2016) approach to rupture behaviour and using the Akkar et al. (2014) GMPE have the largest effect on the calculations, with both assumptions resulting in significant increases to the risk by a factor of 10 or 100, as well as an extension in the number of tolerable risk thresholds exceeded for a given injection volume. The increase in risk observed by assuming the Van der Elst et al. (2016) approach to rupture behaviour is explained by the fact that it leads to larger magnitudes than the base case approach, since it does not constrain the magnitudes of events in line with the volume of fluid injected.
The impact of this assumption is most significant for larger volumes, where the effect of removing the magnitude cap becomes most pronounced. For example, the maximum, 99th percentile, and 95th percentile $M_w > 0$ event simulated for the Van der Elst et al. (2016) approach in the case of 50,000 m³ injected volume were 4.6 $M_w$, 1.7 $M_w$, and 1.1 $M_w$ respectively, whereas the corresponding events simulated for the base case approach were 2.5 $M_w$, 1.5 $M_w$, and 1.1 $M_w$ (see Figure 8 for a comparison of the maximum magnitudes simulated using both source models). Using the Akkar et al. (2014) GMPE leads to increased
risk because it overestimates the ground shaking at PNR, and therefore leads to higher $PGV$ estimates than the Cremen et al. (2019b) GMPE.


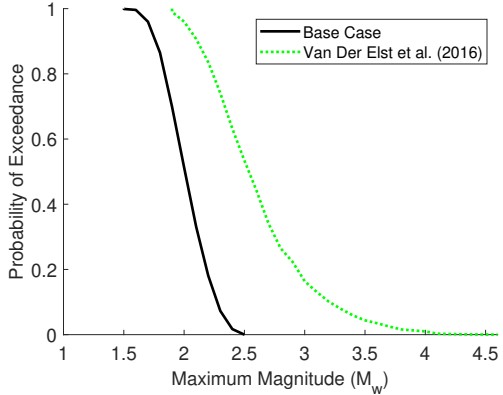

**Figure 8.** Comparing the distribution of maximum magnitudes simulated using the base case approach to rupture behaviour (i.e. magnitudes are capped in proportion to volume injected) and that of Van der Elst et al. (2016) (i.e. magnitudes follow a tectonic Gutenberg-Richter distribution), for 50,000 m$^3$ injected volume.

Accounting for spatial correlation in the ground motions leads to lower probabilities of exceeding most tolerable risk thresholds, across all injection volumes examined. This finding is consistent with previous studies of the effect of spatial dependence on risk (e.g Weatherill et al., 2015). It is explained by the fact that spatial correlation widens the tails of the $PGV$ distribution for a given event, such that small values have a higher probability of being sampled. Note that large $PGV$ values also have a higher probability of being sampled in a spatially correlated portfolio, which should increase the chance of exceeding more severe risk thresholds. However, this is not observed for the exposure model of interest since the effect of spatial clustering (and therefore correlation) is most pronounced for the farthest distances examined (see Figure 1).

Seismicity parameter uncertainty and location uncertainty lead to smaller probabilities of exceeding smaller $PGV$ values, and larger probabilities of exceeding higher $PGV$ values. This implies that the probability distribution of potential ground shaking values widens for both assumptions, which is consistent with expectations for the introduction of uncertainty. The effect of seismicity parameter uncertainty is particularly notable for smaller $PGV$ values. For example, it substantially underpredicts pile driving exceedance probabilities relative to the base case where the parameters are known after-the-fact, which may pose a problem if this threshold is implemented in policies for managing the induced seismicity. Location uncertainty is the least impactful of all assumptions examined. It is most noticeable for larger $PGV$ values, and can lead to slight increases in the probabilities of exceeding the highest thresholds observed for a given injection volume.

The calculations of this section help to increase our understanding of the type of knowledge required to conduct an accurate risk assessment with the proposed framework. We have found that rupture behaviour and event locations are respectively the most and least important pieces of information to constrain for the calculations (at least in the case study of interest).


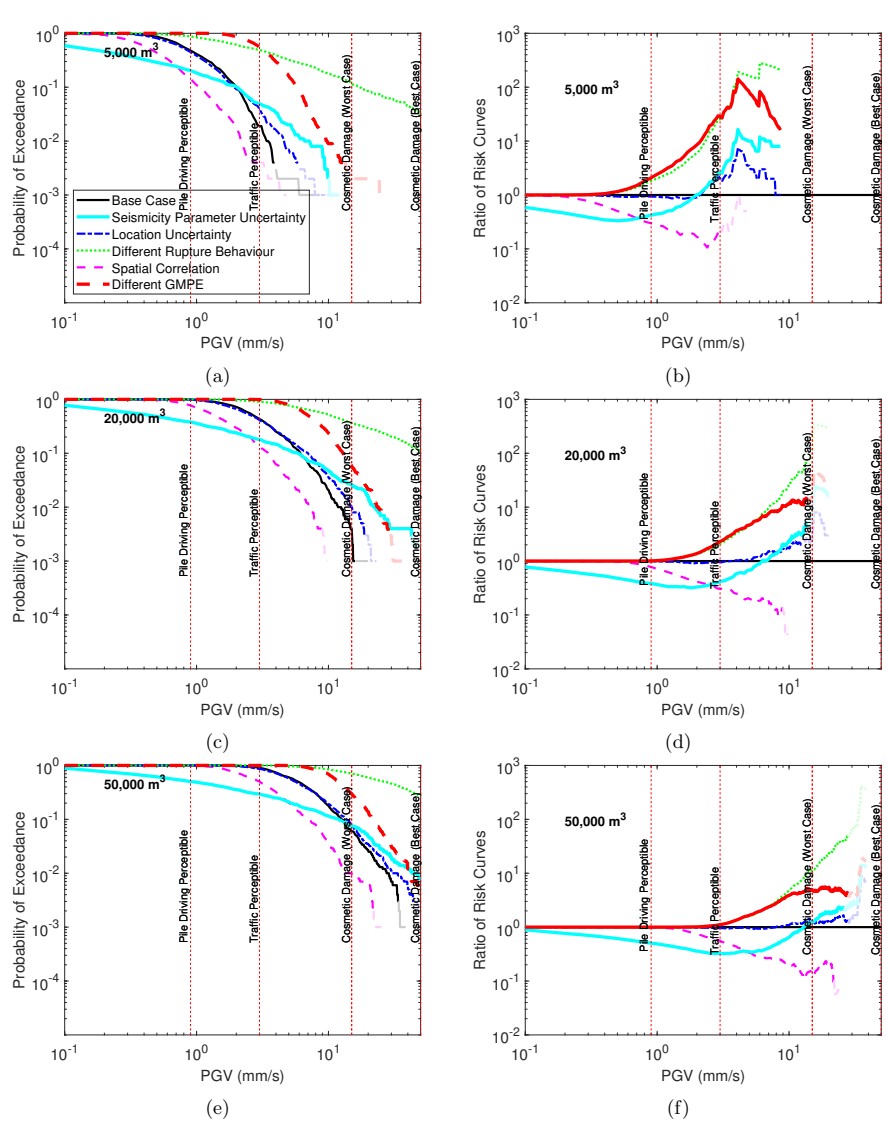

**Figure 9.** Quantifying the impact of alternative modelling assumptions on the probability of exceeding various $PGV$ levels at least once across all considered buildings in the exposure model.





## 6    Implications for Future Policy Design

Based on the recommendations of a hydraulic fracturing review by the Royal Society (Mair et al., 2012), the UK Oil and Gas Authority has implemented a magnitude-based traffic light system (TLS) for the management of induced seismicity related to onshore shale gas development in the country. This TLS allows operations to continue as planned ("green light") when the related induced seismicity is below $M_L = 0$, requires operations to proceed with caution ("amber light") when the seismicity reaches $M_L = 0$ to $M_L = 0.5$, and stipulates a halt in operations ("red light") when seismicity with $M_L \geq 0.5$ occurs.

However, such magnitude-based systems have limited connection to the actual risks associated with the induced seismicity; it is instead the intensity of the ground motions (Bommer et al., 2015), in combination with an exposure model of the surrounding region (Lee et al., 2019), which determine the probability for nuisance or more damaging consequences. The results presented in Section 4.1 of this study could be used to design a more risk-orientated TLS for induced seismicity related to UK hydraulic fracturing, in which the magnitudes corresponding to each level of the system are chosen based on their potential to lead to ground motions that may cause nuisance consequences in the nearby area. For example, an "amber light" may correspond to the lowest magnitude for which there is a non-zero probability of the pile driving threshold being exceeded at any building ($M_w = 1.1$ for PNR from Section 4.1) and a "red light" may correspond to a magnitude just below that for which there is a non-zero chance of cosmetic damage occurring at any building in a worst case scenario ($M_w = 2.1$ for PNR). Similar approaches have been adopted for enhanced-geothermal-induced seismicity (e.g. Bommer et al., 2006; Häring et al., 2008; Ader et al., 2019), although our study has been informed by a more comprehensive analysis of the nearby exposure.

Alternatively, the proposed framework in equation 1 and the results of Section 4.2 could be used to design an injection-volume-based TLS for managing the risk associated with UK hydraulic-fracture-induced seismicity, where each level of the system corresponds to volumes of injected fluid with certain probabilities of causing ground motions that have nuisance potential. For example, an "amber light" could correspond to the first volume for which there is a non-zero probability of the pile driving threshold being exceeded, i.e. 1000 m$^3$ for PNR from Section 4.2, which is roughly equivalent to a quarter of the actual volume injected during PNR-1z operations (Clarke et al., 2019) and a "red light" could correspond to a volume just less than that for which there is a non-zero chance of cosmetic damage occurring in a worst case scenario, i.e. 40,000 m$^3$ for PNR, which is approximately equal to the planned injection volume for PNR-2 (Cuadrilla, 2019).

A significant advantage of this approach over conventional magnitude- or ground motion-based TLSs is that it is proactive rather than reactive, since the injection volume can be controlled ahead of time to avoid a "red light" ever occurring. The findings of Section 5 suggest that a notable amount of *a-priori* information would be needed for such a system to perform accurately, however, particularly related to the rupture behaviour, the seismicity parameters (which can vary greatly even between stages), and the appropriate GMPE for modelling ground motions. The most effective means of implementing the proposed framework would therefore involve real-time updating of the seismicity forecast and the ground motion prediction based on a suite of pre-selected candidate models, following previously proposed approaches for EGS-related seismicity (e.g. Bachmann et al., 2011; Mena et al., 2013; Mignan et al., 2017; Broccardo et al., 2017).





## 7 Possible Limitations

### 7.1 Compatibility of GMPE Predictions with Vibration Thresholds

Values of $PGV$ predicted by a given GMPE may not be strictly compatible with the suggested nuisance vibration thresholds of the proposed framework. This is because GMPEs generally predict either horizontal or vertical amplitude, whereas the thresholds refer to the maximum amplitude across all 3 orthogonal components.

The Cremen et al. (2019b) GMPE used for the framework's application in Section 3 predicts horizontal $PGV$ in terms of the geometric mean across both orthogonal directions. Previous work by Beyer and Bommer (2006) suggests that the maximum

horizontal amplitude - which is expected to be the maximum of all 3 components (e.g. Ghofrani and Atkinson, 2014) - is 15% larger than this value on average. We now investigate the implications on the risk calculations of the expected discrepancy between the GMPE ground shaking amplitudes and the maximum amplitudes. Maximum amplitudes are obtained by scaling the simulated ground motion intensities of Step 4 in the Monte Carlo procedure (Section 3.3) by 1.15.

Figure 10a compares the probability of exceeding $PGV$ values for ground shaking predicted by the GMPE (black lines) and

the equivalent maximum expected amplitude (blue lines), across various injection volumes. It is seen that differences between the two sets of curves are negligible. Figure 10b plots, for 50,000 m³ injected volume, the ratio of the risk curve for the maximum shaking amplitude to that for the shaking predicted by the GMPE (equation 10), and compares it to the equivalent ratio obtained for location uncertainty in Figure 9, which was found to have the least impact on the risk calculations in Section 5. The two sets of ratios are broadly in line across the $PGV$ values examined, which further confirms the insignificant effect of

substituting the GMPE's ground shaking predictions with the maximum ground motion amplitudes expected. We therefore conclude that while there is a discrepancy between the velocities predicted by the GMPE and the definitions of the nuisance risk thresholds implemented, it does not have a significant impact on the risk calculations for the case study of interest.

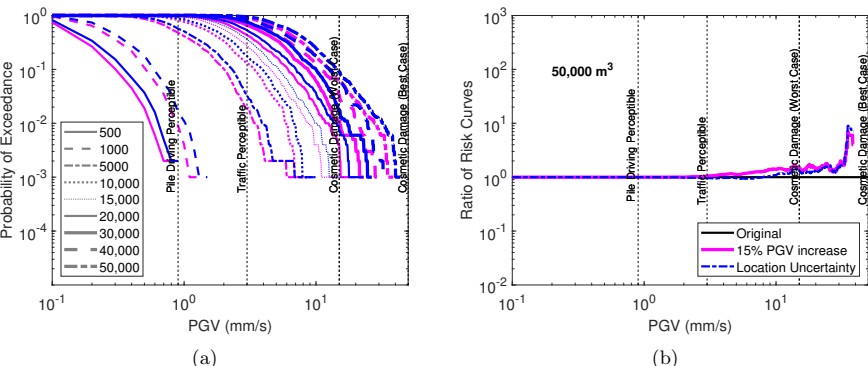

**Figure 10.** Quantifying the impact of increasing $PGV$ predictions by 15% on the probability of exceeding various $PGV$ levels at least once across all considered buildings in the exposure model.





## 7.2  Adequacy of Source Model Assumption

This study has assumed that the rate of earthquakes during hydraulic fracturing is related to the volume of injected fluid, which
has two main limitations: (1) There is no explicit temporal description of seismicity, i.e. the time period of event occurrence is
not considered; and (2) The relationship only applies during the fluid injection phase, such that additional models are needed
to describe postinjection seismicity, e.g. the decay law proposed by Langenbruch and Shapiro (2010). Many other types of
forecast models have been proposed in the literature for injection-induced seismicity that may be more suitable for modelling
earthquake occurrence in our case. These include models in which seismicity is instead proportional to the injection rate (e.g.
Gischig and Wiemer, 2013) and so-called epidemic-type aftershock sequence (ETAS) models (e.g. Hainzl and Ogata, 2005).
Evaluating the performance of various forecast models is outside the scope of this work. The proposed risk framework is
capable of incorporating different types of source models in future studies if necessary however, by appropriately substituting
the "$V_t$" term in equation 1.

## 8  Conclusions

This study has presented a novel framework for assessing some of the consequences of hydraulic-fracture-induced seismic-
ity. The framework explicitly links the volume of fluid injected during operations to the risk of nuisance ground shaking, by
combining statistical forecast models for injection-related seismicity, ground motion prediction equations for hydraulic frac-
turing, exposure models for affected regions, and suggested nuisance risk thresholds adopted from previous studies on human
discomfort to vibrations.
We have demonstrated and validated the proposed modelling approach, using the UK Preston New Road (PNR) shale gas site
and its surrounding area as a test bed. In particular, we showed how the framework can be used to determine event magnitudes
and injection volumes for which prescribed nuisance risk thresholds may be exceeded at buildings nearby the site. For the
specific case study examined, in which seismic events were deterministically located close to the surface projection of the
PNR well stimulated in late 2018, we found that ground motions equivalent in amplitude to that at which pile driving becomes
perceptible may be exceeded in the location of at least one building for event magnitudes equal to or exceeding the current
UK induced seismicity traffic light system "red light" event (i.e. $M_w = 1.1$), or injection volumes $\geq 1000$ m$^3$, while cosmetic
damage may occur in at least one building for $M_w \geq 2.1$ or injection volumes $\geq 40,000$ m$^3$.
We have also examined the sensitivity of the calculations to various modelling assumptions, to better understand the type
of information required for conducting accurate risk assessments with the proposed framework. Implementing two different
models for rupture behaviour (that both aligned reasonably well with observed source data) led to significantly varied risk
estimates in particular. This work therefore highlights the importance of better understanding the physics to quantify likelihoods
of different types of volume-related rupture, i.e. volume-capped moment release versus seismicity that follows a tectonic
Gutenberg-Richter distribution. Use of an appropriate GMPE is also essential for obtaining accurate risk estimates. On the
other hand, we found that constraining event locations would not have a significant effect on the calculations (at least for the
test bed considered in this study).





Finally, we discussed ways in which the proposed modelling approach could contribute to developing risk-informed policies for the management of induced seismicity related to UK shale gas development. For example, we suggested that the framework could be used to design an injection-volume-based traffic light system for induced seismicity based on real-time updating of the model parameters, which would enable injection volumes to be controlled ahead of time to mitigate the probabilities of

causing ground motions with nuisance risk potential. This proactive system could replace the reactive magnitude-based traffic light system currently used in the UK, in which the thresholds do not explicitly account for the associated risks. We expect the findings of this study to be helpful as a decision support tool for stakeholders involved in the regulation of shale gas development in the UK.

*Author contributions.* Both authors conceived and designed the research. GC drafted the written content of the manuscript, which both

authors reviewed. GC performed the calculations. Both authors developed the figures.

*Competing interests.* The authors declare that they have no competing interests.

*Acknowledgements.* This work has been funded by the Natural Environment Research Council (NERC) Grant Number NE/R017956/1 "Evaluation, Quantification and Identification of Pathways and Targets for the assessment of Shale Gas RISK (EQUIPT4RISK)"and the Bristol University Microseismic Projects (BUMPS).



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
