# Peer review of "A Novel Approach to Assessing Nuisance Risk from Seismicity Induced by UK Shale Gas Development, with Implications for Future Policy Design"

_Natural Hazards and Earth System Sciences, 2020_

## Referee Comment (RC1) · Anonymous Referee #1 · 2 Jun 2020

The article "A Novel Approach to Assessing Nuisance Risk from Seismicity Induced by UK Shale Gas Development, with Implications for Future Policy Design" presents an interesting approach for assessing induced seismicity risk from fluid injection operations, with a focus on hydraulic fracturing and shale gas extraction. This approach is primarily based on the volume of fluids injected in the well, combined with ground motion prediction equations and statistical models for earthquake generation, to derive probabilities of exceedance of prescribed ground motion thresholds. The model is initially calibrated and applied to the Preston New Road shale gas site. Then the authors

discuss a number of uncertainties related to the model parameters and evaluate their influence over the derived risk levels. Overall, the article is well-structured and written and clearly seems to be within the scope of Natural Hazards and Earth System Sciences. Therefore, I recommend its publication. However, there are a few points that need some further clarifications and possibly revisions.

The rupture behavior was found to be the most important source of uncertainty in the derived ground motion values. While in the volume-based model the authors explore the uncertainty in the b-values and SEFF in Section 5.1, it would also be interesting to explore the effect of using different shear modulus values for shales that are commonly reported in the literature.

Page 13, Line 277 "b-values values". Remove repetition.

In Fig.10a the authors plot the probabilities of exceeding prescribed PGV values according to the GMPE and its equivalent for the maximum expected amplitude, as magenta and blue color lines, while in Line 459 refer to black lines. Please clarify.

---

## Referee Comment (RC2) · Anonymous Referee #2 · 10 Jul 2020

In this paper the authors propose a new modelling approach which links the volume of fluid injected during operations to the risk of nuisance ground shaking instead of the typical reactive-type magnitude and ground motion-based systems. The paper is well organised and well written and needs only few minor revisions before its publication in this journal. My additional comments to those already provided by the Anonymous Referee #1 are given below:

1) Section 3: I suggest to provide further information about induced seismicity in the shale gas site in Lancashire, including some references.

[Figure]

2) It would be interesting to include in the proposed risk modelling (or discuss at least the inclusion in a future work) the local seismic response and the soil-building resonance effects (e.g., see Gueguen et al., 2002, doi:10.1785/0120000306; Petrovic et al., 2016, doi: 10.1785/0120150326; Gallipoli et al., 2020, doi: 10.1016/j.enggeo.2020.105645), because they may significantly affect the derived ground motion values of your model.

3) Please order multiple citations according to the time of publication. As an example, at page 5, line 121, you should substitute (Ader et al., 2019; Walters et al., 2015) with (Walters et al., 2015; Ader et al., 2019); make the same operation at page 1 (line 15 and line 18), page 6 (line 128), and so on.

―――――――――――――――――

---

## Author Comment (AC1) · 3 Aug 2020

**Response to comments for "A Novel Approach to Assessing Nuisance Risk from Seismicity Induced by UK Shale Gas Development, with Implications for Future Policy Design", by Gemma Cremen and Maximilian J. Werner**

**NHESS-2020-95**

We thank the reviewer for their thoughtful comments, which have improved the quality of the revised manuscript. The reviewer's comments have been numbered and listed below, followed by our responses in italics.

**Reviewer 1:**

1. The rupture behavior was found to be the most important source of uncertainty in the derived ground motion values. While in the volume-based model the authors explore the uncertainty in the b-values and SEFF in Section 5.1, it would also be interesting to explore the effect of using different shear modulus values for shales that are commonly reported in the literature.

   *This is a good point. The authors have implemented this suggestion by also considering shear modulus uncertainty in the seismicity parameter uncertainty analysis of Section 5.1. Shear modulus is now treated as a uniform distribution between 10 and 20 GPa, based on the range of moduli values provided for Bowland Shale (i.e., the formation targeted at Preston New Road in the UK) in:*

   De Pater, C. and Baisch, S.: Geomechanical study of Bowland Shale seismicity, Synthesis report, 57, 2011.

   *Figure 1 compares the exceedance and risk ratio curves (provided in Figure 9 of the paper) when the shear modulus uncertainty is (grey line) and is not (cyan line) accounted for in the seismicity parameter uncertainty analysis. It can be seen that there is no significant difference between the results of the two cases. Therefore, the conclusions about the effect of seismicity parameter uncertainty (discussed in Section 5.6) are unchanged, although the text in Section 5.1 has been updated to describe the additional uncertainty. The updated version of Figure 9 in the paper is provided in Figure 2 below.*

[Figure]

*Figure 1: Comparing the impact of modelling assumptions when shear modulus uncertainty is (grey line) and is not (cyan line) in the seismicity parameter uncertainty analysis.*

[Figure]

Figure 2: Updated Figure 9 of the paper, to reflect the shear modulus uncertainty that is now considered for the seismicity parameter uncertainty case.

2. Page 13, Line 277 "b-values values". Remove repetition.

*Thank you for pointing this out. This error has been fixed in the updated version of the paper.*

3. In Fig.10a the authors plot the probabilities of exceeding prescribed PGV values according to the GMPE and its equivalent for the maximum expected amplitude, as magenta and blue color lines, while in Line 459 refer to black lines. Please clarify.

*Thank you for pointing this out. The "black" description of Line 459 was an error; it has been changed to the correct "magenta" description in the updated version of the paper.*

---

## Author Comment (AC2) · 3 Aug 2020

**Response to comments for "A Novel Approach to Assessing Nuisance Risk from Seismicity Induced by UK Shale Gas Development, with Implications for Future Policy Design", by Gemma Cremen and Maximilian J. Werner**

**NHESS-2020-95**

We thank the reviewer for their thoughtful comments, which have improved the quality of the revised manuscript. The reviewer's comments have been numbered and listed below, followed by our responses in italics.

**Reviewer 2:**

1. Section 3: I suggest to provide further information about induced seismicity in the shale gas site in Lancashire, including some references.

   *We have added several additional references at the start of Section 3:*

   1. *Cremen, G., Werner, M. J., and Baptie, B.: Understanding induced seismicity hazard related to shale gas exploration in the UK, in: SECED 2019 Conference: Earthquake Risk and Engineering towards a Resilient World, 2019a*
   2. *Baptie, B.: Earthquake Seismology 2018/2019, British Geological Survey Open Report OR/19/039, 2019*
   3. *Clarke, H., Verdon, J. P., Kettlety, T., Baird, A., and Kendall, J.-M.: Real-time imaging, forecasting, and management of human-induced seismicity at Preston New Road, Lancashire, England, Seismological Research Letters, 90, 1902–1915, 2019*
   4. *Kettlety, T., Verdon, J., Werner, M., and Kendall, J.: Stress transfer from opening hydraulic fractures controls the distribution of induced seismicity, Journal of Geophysical Research: Solid Earth, 125, e2019JB018 794, 2020*
   5. *Mair, R., Bickle, M., Goodman, D., Koppelman, B., Roberts, J., Selley, R., Shipton, Z., Thomas, H., Walker, A., Woods, E., et al.: Shale gas extraction in the UK: a review of hydraulic fracturing, Royal Society and Royal Academy of Engineering, 2012.*

   *In addition, we have provided further description on the magnitudes of the events (which was already provided in the Introduction) and we have included some background information on induced seismicity in the UK, for context. The first paragraph of Section 3 now reads:*

   *We apply the proposed framework to the region surrounding the Preston New Road (PNR) shale gas site in Lancashire, North West England, where hydraulic fracturing operations in late 2018 (at PNR-1z well) and mid 2019 (at PNR-2 well) resulted in 29 ML > 0 events with maximum magnitude ML = 2.9, eight of which were felt nearby (e.g., Baptie, 2019; Cremen et al., 2019a, b; Clarke et al., 2019; Kettlety et al., 2020). Shale gas development is a relatively new source of induced seismicity in the UK (Clarke et al., 2014), and the PNR activities are the only hydraulic fracturing operations to take place onshore in the country between a 2012 government-ordered investigation into the related risks (Mair et al., 2012) and the hydraulic fracturing moratorium imposed in England in November 2019. For the purposes of this application, we assume that seismicity is produced from a point source 2 km deep (i.e. ns = 1 in equation 1), at a respective*

*latitude and longitude of 53.7873◦ North and 2.9511◦ West. This location corresponds to the approximate depth of the Bowland shale targeted by the operation and the surface coordinates of the PNR-1z well, according to the 2018 hydraulic fracture plan of the operator (Cuadrilla, 2017)*

2. It would be interesting to include in the proposed risk modelling (or discuss at least the inclusion in a future work) the local seismic response and the soil-building resonance effects (e.g., see Gueguen et al., 2002, doi:10.1785/0120000306; Petrovic et al., 2016, doi: 10.1785/0120150326; Gallipoli et al., 2020, doi: 10.1016/j.enggeo.2020.105645), because they may significantly affect the derived ground motion values of your model.

   *Thank you for this suggestion. While we appreciate that soil-structure interaction can modify structural performance (either favourably or unfavourably), it is recognised that the consideration of soil-structure interaction in large-scale risk analyses (such as this study, which includes > 4000 buildings) can be a challenging and time-consuming process (Silva et al., 2020). An alternative proxy method of accounting for these effects is the consideration of secondary factors (e.g., age) that aggravate or attenuate the seismic vulnerability (Silva et al., 2020). We have already implicitly accounted for these factors in our analysis by considering both "worst case" (i.e. weak structure) and "best case" (i.e. strong structure) scenarios for cosmetic damage.*

   Silva, V., Akkar, S., Baker, J., Bazzurro, P., Castro, J. M., Crowley, H., Dolsek, M., Galasso, C., Lagomarsino, S., Monteiro, R., et al.: Current challenges and future trends in analytical fragility and vulnerability modeling, Earthquake Spectra, 35, 1927–1952, 2019.

3. Please order multiple citations according to the time of publication. As an example, at page 5, line 121, you should substitute (Ader et al., 2019; Walters et al., 2015) with (Walters et al., 2015; Ader et al., 2019); make the same operation at page 1 (line 15 and line 18), page 6 (line 128), and so on.

   *We have implemented this suggestion in the updated version of the paper.*